# Adjacent Image Augmentation and Its Framework for Self-Supervised Learning in Anomaly Detection

**DOI:** 10.3390/s24175616

**Published:** 2024-08-29

**Authors:** Gi Seung Kwon, Yong Suk Choi

**Affiliations:** Department of Computer Science, Hanyang University, Seoul 04763, Republic of Korea; kwonga@hanyang.ac.kr

**Keywords:** anomaly detection, automatic optical inspection, representation learning

## Abstract

Anomaly detection has gained significant attention with the advancements in deep neural networks. Effective training requires both normal and anomalous data, but this often leads to a class imbalance, as anomalous data is scarce. Traditional augmentation methods struggle to maintain the correlation between anomalous patterns and their surroundings. To address this, we propose an adjacent augmentation technique that generates synthetic anomaly images, preserving object shapes while distorting contours to enhance correlation. Experimental results show that adjacent augmentation captures high-quality anomaly features, achieving superior AU-ROC and AU-PR scores compared to existing methods. Additionally, our technique produces synthetic normal images, aiding in learning detailed normal data features and reducing sensitivity to minor variations. Our framework considers all training images within a batch as positive pairs, pairing them with synthetic normal images as positive pairs and with synthetic anomaly images as negative pairs. This compensates for the lack of anomalous features and effectively distinguishes between normal and anomalous features, mitigating class imbalance. Using the ResNet50 network, our model achieved perfect AU-ROC and AU-PR scores of 100% in the bottle category of the MVTec-AD dataset. We are also investigating the relationship between anomalous pattern size and detection performance.

## 1. Introduction

Anomaly detection is a critical task that involves identifying data patterns that deviate significantly from the norm [1,2,3]. This process is essential across various domains such as manufacturing quality inspection [4], medical diagnostics [5], cybersecurity [6], financial monitoring [7], CCTV surveillance [8], and autonomous driving [9]. Typical anomaly detection methods leverage deep learning to map normal data features into a latent space, thereby creating a distribution of normal data. Anomalies are detected by comparing the features of input data against this distribution. Despite its importance, anomaly detection faces significant challenges, primarily due to the class imbalance between normal and anomalous data. Anomalies are rare compared to the vast amount of normal data, making it difficult for models to learn to detect them effectively. Recent research has focused on self-supervised learning techniques, which can help mitigate class imbalance by generating synthetic anomaly data [10,11,12].

One notable approach in self-supervised learning is the use of various augmentation techniques. For instance, CutPaste augmentation involves cutting a rectangular patch from a training image and randomly pasting it back into the original image [13]. This technique introduces anomalies by disrupting normal patterns. Another method, SmoothBlend augmentation, entails cutting a small round patch, applying color jitter, and reinserting it into the image [14]. This method aids in detecting small defects by creating more challenging patterns for the model to learn.

In recent years, contrastive learning frameworks such as SimCLR and SimSiam have gained significant traction [15,16,17,18]. These methods generate two samples by applying different augmentations to the same training image. The SimCLR method designates the two generated images as positive pairs to each other and as negative pairs to images generated from other training data within the same batch. In contrast, the SimSiam method passes the two augmented images through an encoder to create vectors, with only one vector passing through a projection. The vector that passes through both the encoder and projection is then designated as a positive pair with the vector that passes only through the encoder. However, these methods do not designate the training data within the same batch as positive pairs to each other, which limits their ability to effectively learn the nuances of normal images.

To enhance anomaly detection, we propose an adjacent augmentation technique that generates synthetic anomaly images by preserving object shapes and distorting the contours of specified regions. There are three methods of adjacent augmentation for generating synthetic anomaly images: Mosaic, Liquify, and Mosiquify. The Mosaic method reduces the resolution of a selected area and applies color jitter, producing defects that appear more natural. The Liquify method distorts contours to mimic real-world defects like scratches and sagging. The Mosiquify method combines both Mosaic and Liquify augmentations to generate even more realistic anomalies. Additionally, we introduce the Strong Overall and Wake Overall methods for augmenting synthetic normal images. By applying these synthetic images and an anomaly detection benchmark dataset [14,19,20] to our framework, we establish positive pairs between the training images within each batch, and between the training images and synthetic normal images, and negative pairs between the training images and synthetic anomaly images [15,21]. This approach not only helps mitigate class imbalance but also improves the model’s ability to differentiate between normal and anomalous data. Table 1 demonstrates that our proposed augmentation method does not show a significant difference in speed compared to previous augmentation techniques.

Our main contributions can be summarized as follows:We propose novel augmentation techniques and a framework for self-supervised learning aimed at addressing class imbalance in anomaly detection.Our adjacent augmentations generate synthetic anomalies with realistic contour distortions, enhancing the model’s learning process.We develop a contrastive learning framework that leverages characteristics from anomaly detection benchmark datasets, improving the overall effectiveness of anomaly detection models.

## 2. Related Work

### 2.1. MVTec-AD Dataset

The MVTec-AD dataset is a benchmark for anomaly detection, specifically designed for the precise inspection of defects in industrial manufacturing [19]. This dataset includes five texture categories and ten object categories, addressing limitations in the scope of previous anomaly detection datasets. The training set comprises 3629 normal images, while the test set contains 1,725 normal images and a mix of anomaly images. Despite the increased dataset size, the issue of class imbalance persists, with significantly fewer anomaly images. All images are captured using high-resolution RGB sensors, and the anomaly images accurately reflect real-world defects. In the texture categories, images exhibit repeating patterns, whereas object category images are captured in specific locations. Our adjacent framework leverages the fact that all training data in this dataset consist of normal images. Figure 1 shows samples from the MVTec-AD dataset. Table 2 provides a description of the MVTec-AD dataset.

### 2.2. Representative Anomaly Detection

Semi-supervised learning techniques for one-class anomaly detection leverage the feature distribution of normal data to identify anomalies. During training, the model encodes normal data features into a latent space, establishing a distribution that represents normalcy. At inference, the model classifies input data as normal if its features fall within the decision boundary of the normal data distribution. Conversely, if the input data features lie outside this boundary, the data are classified as anomalous [22,23]. Figure 2 illustrates the process of this method.

Autoencoder-based methods perform anomaly detection by reconstructing compressed input data as normal data. During the training phase, the model learns by repeatedly compressing and reconstructing normal data. In the inference phase, the model calculates the reconstruction error between the input data and the reconstructed data. Since the autoencoder reconstructs normal data well, the error is low, and the model classifies it as normal. Conversely, the autoencoder does not reconstruct anomaly data well, resulting in a high error, and the model classifies it as anomalous [24]. Figure 3 illustrates the process of this method.

Finally, feature matching methods detect anomalies by comparing the features of normal data with those of input data. Normal images are divided into small patches, with key features stored in memory. The model calculates the similarity between the input image features and the stored normal features. If the input image features significantly deviate from the stored normal features, the image is classified as anomalous [25,26,27]. Figure 4 illustrates the process of this method.

### 2.3. Class Imbalance

The anomaly detection methods discussed in Section 2.2 typically include only normal data for training due to the class imbalance problem [14,28]. This imbalance arises when normal data significantly outnumbers anomaly data. In a latent space with class imbalance, the feature distribution of normal data dominates, biasing input data towards being classified as normal. This bias can negatively impact anomaly detection performance, necessitating strategies to mitigate class imbalance. Figure 5 visualizes the class imbalance.

### 2.4. SimCLR

Our adjacent framework draws inspiration from the SimCLR framework [15], a contrastive learning method that embeds images into a latent space where positive pairs are closer together and negative pairs are farther apart [29]. SimCLR effectively extracts visual representations through unsupervised learning by generating two differently augmented versions of each training image and treating them as positive pairs, while all other images in the batch are treated as negative pairs [15]. We reference characteristics from benchmark training datasets [14,19,20] to slightly modify the concepts of the SimCLR framework. Our adjacent framework generates two synthetic normal images and one synthetic anomaly image from each training image. The training image and synthetic normal images are set as positive pairs, while each training image and synthetic anomaly image is set as a negative pair. Additionally, all training images within the batch are treated as positive pairs, helping to establish a robust normal image distribution. This framework enhances the learning of distinctions between normal and anomalous images and employs synthetic anomaly images to address class imbalance. Figure 6 compares our framework with the SimCLR framework.

## 3. Methods

This chapter outlines the methods used to generate synthetic data through adjacent augmentations. Our approach involves augmenting training images to create synthetic normal and synthetic anomaly images. We employ the Strong Overall and Weak Overall methods for generating synthetic normal images and the Mosaic, Liquify, and Mosiquify methods for generating synthetic anomaly images. These synthetic anomaly images closely resemble real defects, helping to address class imbalance. Additionally, we discuss previous augmentation methods, such as CutPaste [13] and SmoothBlend [14], and how adjacent augmentation synthesizes anomalous patterns. The final section details our adjacent framework, which integrates synthetic images for enhanced anomaly detection.

### 3.1. Augmentation

This section describes the augmentation techniques used in our framework and those from previous work. The first two methods involve augmentations that generate positive samples, while the remaining methods involve augmentations that generate negative samples.

Our augmentation methods are based on their effectiveness in generating synthetic anomaly images that closely resemble real-world defects. These methods introduce realistic variations that challenge the model’s ability to distinguish between normal and anomalous data, which is crucial for enhancing anomaly detection performance. Additionally, these techniques allow us to simulate a wide range of defects, addressing the class imbalance issue by providing diverse and realistic anomaly samples. Furthermore, these methods are particularly effective because they exploit the strong correlation between anomalous patterns and surrounding pixels, enabling more effective learning. We believe that these methods strengthen our framework by improving the model’s robustness and generalization capabilities.

#### 3.1.1. Weak Overall

In industrial manufacturing, images are captured individually under varying conditions of lighting, angle, and position, resulting in slight differences. To reduce sensitivity to these minor variations, we use the Weak Overall augmentation from the Spot-the-Difference method. This augmentation helps the model better classify normal images despite these small variations [14]. Figure 7 shows a Weak Overall sample.

Algorithm to generate Weak Overall samples:The first step is to crop the anchor from 90% to 100% and then resize it to the size of the anchor.The second step is to adjust the brightness, contrast, saturation, and hue properties of the anchor to random values between 0% and 10%.The next step is to apply a Gaussian blur with a kernel size of 5 by 5 and a sigma value between 0.1 and 0.3.The final step is to apply a horizontal flip with random probabilities.

#### 3.1.2. Strong Overall

Normal images in industrial manufacturing typically have consistent shapes and contours. To detect small anomalies, the model must analyze detailed features of these images. The Strong Overall augmentation focuses on learning the intricate details of normal images, aiding in the detection of subtle anomalies. Figure 8 shows a Strong Overall sample.

Algorithm to generate Strong Overall samples:The first step is to crop the anchor to a random size and then resize it to the size of the anchor.The second step is to apply horizontal flipping with random probabilities.The next step is to adjust the brightness, contrast, and saturation properties of the anchor to random values between 0% and 80%, and the hue to random values between 0% and 20%.The random grayscale method converts images to black and white with a 20% probability.The final step is to apply a Gaussian blur using a kernel with a size of 10% of the anchor.

#### 3.1.3. CutPaste

CutPaste augmentation involves cutting a square patch from a training image and pasting it back onto the original image [13]. This augmentation distorts the continuous pattern, teaching the model to recognize such disruptions as anomalies. The CutPaste method is effective in highlighting discontinuous patterns indicative of anomalies. Figure 9 shows a CutPaste sample.

Algorithm to generate CutPaste samples:The first step is to apply the Weak Overall augmentation.The second step is to set the size ratio of the patch to 2% to 15% and the aspect ratio to 0.3 to 3.The third step is to cut out the square patch from the anchor to the specified size.The final step is to paste the patch into a random location in the original image.

#### 3.1.4. SmoothBlend

SmoothBlend augmentation cuts a small, round patch from a training image and pastes it onto the original image, distorting its continuous pattern. This augmentation helps the model learn to identify small defects by focusing on these local distortions [14]. Figure 10 shows a SmoothBlend sample.

Algorithm to generate SmoothBlend samples:The first step is to apply the Weak Overall augmentation.The second step is to set the size ratio of the patch to 0.5% to 1%, and the aspect ratio to 0.3 to 3.The third step is to cut out the round patch from the anchor to the specified size.The fourth step is to apply random contrast up to 100% to the patch, random saturation up to 100%, and random color jittering up to 50%.The final step is to alpha blend the original image and its patch.

#### 3.1.5. Mosaic

Drawing inspiration from the SmoothBlend technique, Mosaic augmentation modifies the resolution and color within a specified circular region, rather than employing a cut-and-paste approach. This augmentation introduces subtle, natural-looking anomalies that challenge the model by distorting patterns in a manner highly pertinent to the surrounding pixels. Figure 11 shows a Mosaic sample.

Algorithm to generate Mosaic samples:The first step is to apply the Weak Overall augmentation.The second step is to set the size ratio of the round area to be converted to 0.5% to 1%, and the aspect ratio to 1.The third step is to reduce the specified area to the rate of ζ and restore it to its original size.The fourth step is to apply random brightness up to 50%, random contrast up to 50%, random saturation up to 50%, and random color jittering up to 20%.The final step is to alpha blend the original image and the converted area.

#### 3.1.6. Liquify

Liquify augmentation distorts image contours by displacing random points, thereby generating patterns reminiscent of liquid flow. This technique aids the model in learning to classify distorted contour patterns, effectively simulating natural defects such as scratches and sagging. Figure 12 shows a Liquify sample.

Algorithm to generate Liquify samples:The first step is to apply the Weak Overall augmentation.The second step is to assign a random point to the image.The third step specifies each coordinate of the four triangles centered around the designated point.The fourth step moves the specified point to a random location at a distance of the image size × (1/η)%.In the final step, four triangles move as the point moves, creating contour distortion.

#### 3.1.7. Mosiquify

Mosiquify augmentation synergistically combines the effects of Liquify and Mosaic augmentations, thereby distorting contour, resolution, and color. This technique introduces complex and varied anomalies, facilitating the model’s ability to recognize a wide range of anomalous features. Figure 13 shows a Mosiquify sample.

Algorithm to generate Mosiquify samples:The first step is to apply the Weak Overall augmentation.The second step is to apply the Mosaic (ζ = 20) augmentation.The final step is to apply the Liquify (η = 0.05) augmentation.

### 3.2. Adjacent Framework

The adjacent framework encompasses both image augmentation and the learning process. Traditional anomaly detection contrastive learning frameworks typically employ straightforward augmentations and a single loss function. In contrast, our framework introduces novel augmentations and utilizes two distinct loss functions to enhance the learning of features from both normal and anomalous data. Furthermore, unlike previous frameworks that focus solely on augmenting anomalous images, our framework applies augmentation to both normal and anomalous images. Finally, our contrastive learning framework maximizes the embedding distance between normal and anomalous data by leveraging NCE loss and cosine similarity loss. Figure 14 shows the augmentations used in the adjacent framework. The detailed process of the adjacent framework is shown in Figure 15.

Our contrastive learning framework focuses on learning effective representations by embedding similar (positive) samples closer together in a latent space, while pushing dissimilar (negative) samples farther apart. Specifically, we augment each anchor image with both positive samples (such as Weak Overall and Strong Overall augmentations) and negative samples (synthetic anomaly images). We utilize losses like NCE loss and cosine similarity loss to ensure that positive pairs are closely aligned, and negative pairs are distinct within the feature space. This approach not only improves the model’s ability to distinguish between normal and anomalous data but also addresses class imbalance by leveraging synthetic anomaly images.

Our adjacent framework leverages synthetic images in conjunction with the anomaly detection benchmark training dataset. This framework employs a self-supervised learning method known as contrastive learning. In this approach, artificial labels are generated from the data to train the model, enabling the effective utilization of unlabeled data. The framework enhances similarity between positive pairs while reducing similarity between negative pairs. The contrastive learning loss function aims to maximize the similarity of positive sample pairs by minimizing their distance. Conversely, for negative sample pairs, the objective is to maximize the distance, thereby minimizing their similarity. By optimizing this loss function, the model learns meaningful representations, effectively distinguishing between similar and dissimilar data points [15].
(1)$$L_{Positive}\left(x_{i},\;x_{j}\right)=-\log\frac{\exp\left(Similarity(x_{i},\;x_{j})/\tau \right)}{\sum_{k=1}^{N}1_{k\neq i}\exp\left(Similarity(x_{i},\;x_{k})/\tau \right)}\;$$
(2)$$m:Minimum\;distance\;between\;negative\;pairs\;L_{Negative}\left(x_{i},\;x_{j}\right)=-\log\frac{\exp\left(m-Similarity(x_{i},\;x_{j})/\tau \right)}{\sum_{k=1}^{N}1_{k\neq i}\exp\left(m-Similarity(x_{i},\;x_{k})/\tau \right)}\;$$

In our framework, training images paired with synthetic normal images are designated as positive pairs, while training images paired with synthetic anomaly images are designated as negative pairs. Furthermore, all training images within the batch are considered positive pairs. We employ InfoNCE loss and cosine similarity loss to train the model. The InfoNCE loss function encourages the model to draw the anchor and positive pair representations closer together while pushing the anchor and negative pair representations further apart [14,21]. The cosine similarity loss function maximizes the similarity between positive pairs and minimizes the similarity between negative pairs. Generating synthetic normal data assists the model in learning detailed features. Anchors and strong overall samples are set as positive pairs, with InfoNCE loss bringing them closer together, thereby reducing sensitivity to environmental changes. Anchors and weak overall samples are also set as positive pairs, while synthetic anomaly data are designated as negative pairs, with cosine similarity loss managing these relationships [14,30].
(3)$$L_{NCE}\left(x_{i},{\hat{x}}_{i}\right)=-\log\frac{\exp\left(z_{i}\cdot {\hat{z}}_{i}/\tau \right)}{\exp\left(z_{i}\cdot {\hat{z}}_{j}/\tau \right)+\sum_{j=1}^{N}1_{j\neq i}\exp\left(z_{i}\cdot {\hat{z}}_{j}/\tau \right)}$$
(4)$$Cosine\;Similarity=\frac{A\cdot B}{\left|\left|A\right|\right|\left|\left|B\right|\right|}=\frac{\sum_{i=1}^{n}A_{i}B_{i}}{\sqrt{\sum_{i=1}^{n}A_{i}^{2}}\sqrt{\sum_{i=1}^{n}B_{i}^{2}}}$$

Each training image serves as an anchor, generating Strong Overall, Weak Overall, and negative samples. The framework ensures that anchors, training data, and Strong Overall samples are closely embedded, while anchors and synthetic anomaly images are kept distinct. In summary, our adjacent augmentations and framework generate synthetic images and utilize them for contrastive learning. The training image passes through the encoder to become a representation, which then undergoes projection and normalization. This process enables the model to effectively learn the differences between normal and anomalous images, addressing class imbalance and enhancing anomaly detection performance.

Algorithm 1 summarizes the proposed method. Algorithm 1 augments one anchor with three samples (Weak Overall sample, Strong Overall sample, and Negative sample). The anchor and positive samples are embedded closer together using NCE loss and cosine similarity loss. The anchor and negative sample are embedded farther apart using cosine similarity loss.

**Algorithm 1** Adjacent Framework’s main learning algorithm**Input**: batch size N, constant τ, structure of f,g,Τ,δ.**for** sampled minibatch xkk=1N **do** **for all**
k∈1,…,N **do** draw three augmentation functions t~Τ,t′~Τ,t″~Τ  # anchor  x~4k−3=xk
  h4k−3=f(x~4k−3)
  z4k−3=g(h4k−3)
  # the first augmentation(Weak Overall-positive)  x~4k−2=t(xk)
  h4k−2=f(x~4k−2)
  z4k−2=g(h4k−2)
  # the second augmentation(Strong Overall-positive)  x~4k−1=t′(xk)
  h4k−1=f(x~3k−1)
  z4k−1=g(h3k−1)
  # the third augmentation(Liquify-negative)  x~4k=t″(xk)
  h4k=f(x~3k)
  z4k=g(h3k)
  **for all**
$i,j,m,n\in \left\{1,\ldots ,4N\right\}\;$**do**  **define**
$l_{nce}\left(i,j\right)\;as\;l_{nce}\left(i,j\right)=-log\frac{\exp\left(z_{i}\cdot z_{j}/\tau \right)}{\exp\left(z_{i}\cdot z_{j}/\tau \right)+\sum_{k=1}^{2N}1_{\left[k\neq i\right]}\exp\left(z_{i}\cdot z_{j}/\tau \right)}$  **define**
$l_{cosine}\left(i,m,n\right)\;as\;l_{cosine}\left(i,m,n\right)=-\frac{z_{i}\cdot z_{m}}{\left|\left|z_{i}\right|\right|\left|\left|z_{m}\right|\right|}+\frac{z_{i}\cdot z_{n}}{\left|\left|z_{i}\right|\right|\left|\left|z_{n}\right|\right|}$  L= $l_{nce}\left(4k-3,\;4k-2\right)+\delta \times l_{cosine}(4k-3,\;4k-1,\;4k)$  update networks f and g to minimize L  **end for** **end for** return encoder network f(∙), and throw away g(∙)

## 4. Experiments

In our experiments, we employed a ResNet50 backbone network pre-trained on the ImageNet 1K dataset, with output classes designated as normal and anomaly. Input images were resized to 512 × 512 pixels and subsequently augmented. We used the NVIDIA GeForce RTX 2080 Ti GPU for computational efficiency. The hyperparameters were configured identically to those used in the Spot-the-Difference experiments [14]. The Adam optimizer was utilized with a learning rate of 0.0001 and a weight decay of 0.00003. Additionally, we applied the Cosine Annealing Learning Rate method to gradually decrease the optimizer’s learning rate following a cosine curve. The batch size was set to 16, and the temperature parameter was set to 0.1. Training was conducted for 800 epochs, with model evaluation performed after each epoch using the test dataset. We saved the model when the accuracy, AU-ROC curve, and AU-PR curve achieved their highest values. The model was trained on a single category at a time, ensuring a one-to-one correspondence between the category and the model. We conducted experiments under these conditions and compared the results by applying different augmentations within the adjacent framework. We report the maximum Area Under the Receiver Operating Characteristics (AU-ROC) and Area Under the Precision-Recall (AU-PR) curves for each category in the MVTec-AD dataset. Finally, ζ is a parameter that controls the size of mosaic anomaly patterns, and η is a parameter that controls the size of Liquify anomaly patterns. Figure 16 shows images of real-world defects alongside images generated using adjacent augmentation.

Additionally, Appendix A provides explanations and experimental results that were not included in the main paper. Figure A1 shows the change in loss over 500 epochs of training. Figure A2 shows the changes in accuracy curves over 500 epochs of training. Figure A3 shows the changes in ROC curves over 500 epochs of training. Figure A4 provides a brief explanation of the evaluation metrics we use.

Table 3 illustrates the anomaly detection performance of models trained with Liquify augmentation within the adjacent framework. Our framework designates training data within the batch as positive pairs, thereby standardizing the features of normal data. We compared the performance of various augmentations based on the adjacent framework with those of the SimCLR framework.

Table 4 compares synthetic anomaly images generated by previous augmentation methods with those generated by highly correlated adjacent augmentation. Parameters ζ and η indicate the degree of transformation applied by the adjacent augmentation. We present the maximum AU-ROC and AU-PR for 10 categories in the MVTec-AD dataset. Table 5 provides an ablation study on the impact of excluding synthetic anomaly data as negative samples within the adjacent framework. Synthetic anomaly images generated by adjacent augmentations have contours like real-world defects. By using these synthetic anomaly images as negative samples, the model learns improved anomaly features. The ‘none’ column represents learning without generating negative samples from the adjacent framework. Figure 17 illustrates the size of the Liquify pattern according to the parameter η, which controls the distance that a point moves. Table 6 shows the relationship between anomaly detection performance and the size of Liquify patterns. We provide maximum AU-ROC and AU-PR for 15 categories in the MVTec-AD dataset. Finally, we compared our method with various anomaly detection algorithms. Our adjacent framework, incorporating synthetic images and contrastive learning, demonstrated superior performance across multiple categories, highlighting its effectiveness in addressing class imbalance and improving anomaly detection. Table 7 shows the results of applying our method to the VisA dataset. Table 8 compares our proposed method with various anomaly detection approaches.

## 5. Discussion

### 5.1. Summary of Findings

In this paper, we introduce the adjacent augmentation technique and its framework to address the persistent challenge in anomaly detection. Our method integrates image augmentation with a learning framework to improve the recognition and identification of anomalous patterns. Adjacent augmentation addresses class imbalance by generating high-quality anomalous image features that retain shape while distorting contours, thus enhancing correlation with normal images. The adjacent framework standardizes the distribution of normal features by treating all training data within a batch as positive pairs and effectively learns the distinctions between normal and anomalous features using synthetic images. In other words, our augmentation methods simulate real-world defect patterns by introducing controlled distortions that resemble actual anomalies. For instance, as shown in Figure 15, positive samples generated through adjacent augmentations are embedded closer to the anchor using NCE loss and cosine similarity loss. In contrast, negative samples generated by the Mosaic, Liquify, and Mosiquify methods are embedded farther from the anchor using cosine similarity loss. The advantage of our generation method lies in its ability to produce a wide range of realistic anomalies that closely mimic real-world defects. This enhances the model’s ability to distinguish between normal and anomalous data, making it more robust compared to other generation methods that might only focus on simpler or less varied synthetic defects.

### 5.2. Comparison with Existing Methods

CutPaste and SmoothBlend are effective in generating synthetic anomalies, but they primarily rely on simple cut-and-paste operations or blending techniques, which may not fully capture the complexity of real-world defects. These methods often struggle to simulate the intricate anomaly patterns found in diverse industrial settings, and they can sometimes introduce unrealistic artifacts that hinder the model’s generalization ability.

In contrast, our proposed methods—Mosaic, Liquify, and Mosiquify—create more complex and realistic synthetic anomalies that better resemble real-world defects. By focusing on both local and global image distortions, our methods effectively simulate a broader range of anomaly types. Furthermore, our adjacent framework leverages the correlation between anomalous patterns and surrounding pixels, leading to more robust learning and better detection performance.

Building on this, traditional approaches like CutPaste and SmoothBlend often suffer from low correlation between the anomalous pattern and its surrounding area, which can result in an ineffective learning of anomalies. In contrast, our adjacent augmentation technique generates highly correlated anomalous patterns, facilitating more effective integration into normal images. This was evidenced by our experiments, which demonstrated the significant impact of these highly correlated patterns on anomaly detection performance. As a result, our method outperformed existing techniques such as CutPaste and SPD, significantly improving AU-ROC and AU-PR scores across various categories in the MVTec-AD dataset.

### 5.3. Impact of Deep Learning Architecture

The effectiveness of anomaly detection is significantly influenced by the choice of deep learning architecture. In our study, we employed the ResNet50 backbone network, renowned for its capability to learn complex representations in image data. The residual connections in ResNet50 mitigate the vanishing gradient problem and enable the training of deeper networks, thereby capturing intricate patterns in the data. Furthermore, our framework utilizes contrastive learning to enhance the model’s ability to learn meaningful representations. By maximizing the similarity between positive pairs and minimizing it between negative pairs, the model can effectively discriminate between similar and dissimilar data points. This approach aligns with advancements in self-supervised learning, which have demonstrated superior performance across various tasks.

### 5.4. Limitations

Our adjacent augmentation method is currently focused on image-based anomaly detection, thereby limiting its applicability to other types of datasets, such as text, time-series, or video data. Achieving comparable performance on these data types may necessitate additional research and the development of specialized augmentation techniques. Furthermore, feature matching-based anomaly detection methods can offer more accurate detection through advanced feature extraction and matching algorithms. However, our method does not fully integrate these complex matching techniques, which may constrain its ability to detect subtle differences in high-dimensional feature spaces. Specifically, our approach may not perform optimally in scenarios where detecting subtle anomalous patterns that closely resemble normal patterns is critical. Considering these limitations, our research introduces a novel approach to image-based anomaly detection, but further investigation and improvements are required to extend its applicability to diverse data types and more complex anomaly detection scenarios. Future work will focus on overcoming these limitations and enhancing our method to increase its applicability across various domains.

## 6. Conclusions

Our adjacent augmentation method enhances anomaly detection performance by generating high-quality synthetic anomalies that are closely correlated with their surroundings. Through extensive experiments, we demonstrated the effectiveness of our approach in alleviating class imbalance and improving model performance. By leveraging contrastive learning and robust deep learning architectures, our framework makes significant contributions to the field of anomaly detection. The potential applications of our method are vast, offering improved reliability and accuracy in various industrial contexts.

## Figures and Tables

**Figure 1 sensors-24-05616-f001:**
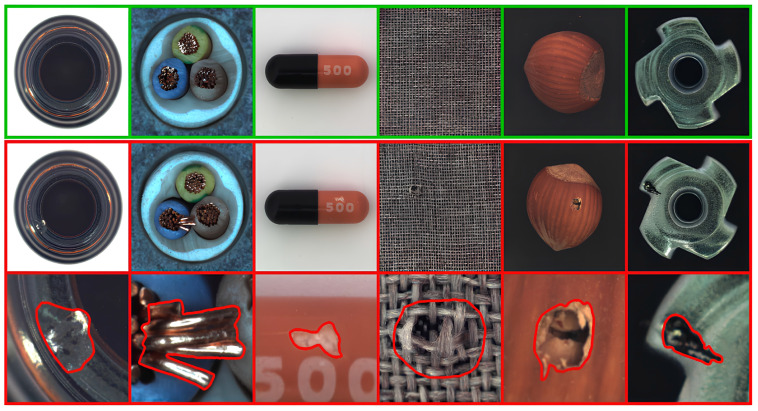
Images from the MVTec-AD dataset. This dataset comprises object and texture classes. Normal images feature a green border, while anomaly images are outlined in red. Defects in this dataset are indicated by a red border.

**Figure 2 sensors-24-05616-f002:**
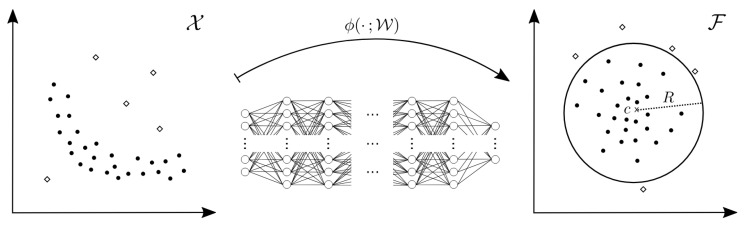
Utilizing deep one-class classification for anomaly detection. This algorithm determines the normalcy of input data by assessing whether they resides within a hypersphere formed by normal data. The figure illustrates the process of constructing such a hypersphere using a neural net1work to discern the features characteristic of normal data.

**Figure 3 sensors-24-05616-f003:**
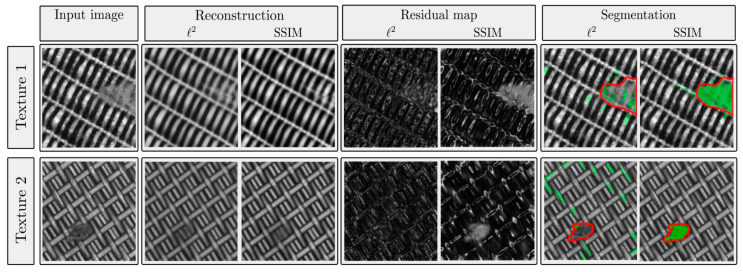
This figure compares $l^{2}$-autoencoder and SSIM-autoencoder for anomaly detection using autoencoders. An autoencoder trained on normal data compresses input fabric textures and then reconstructs them as normal fabric textures. The $l^{2}$-autoencoder removes defects during reconstruction, while the SSIM-autoencoder retains defects. Therefore, the SSIM-autoencoder shows better anomaly detection performance than the $l^{2}$-autoencoder.

**Figure 4 sensors-24-05616-f004:**
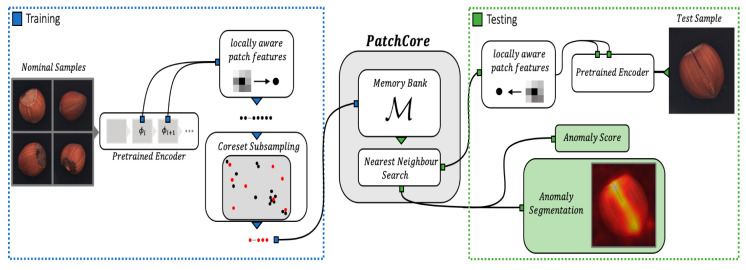
Utilizing memory bank for anomaly detection. The memory bank retains features extracted from normal patches. The model then compares the features of the input image with those stored in the memory bank. If there’s at least one discrepancy between the input patches and the stored normal patches, the model classifies the input image as an anomaly.

**Figure 5 sensors-24-05616-f005:**
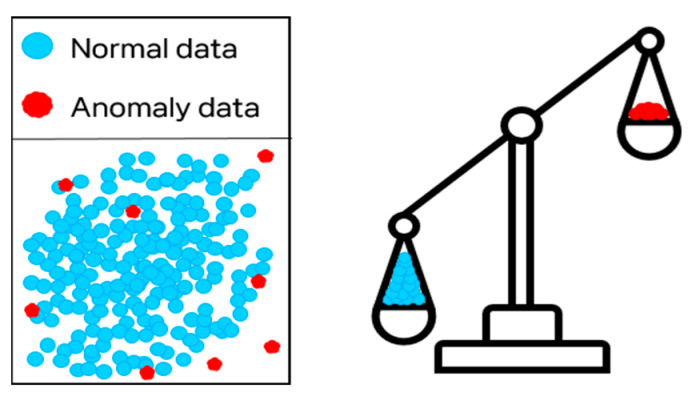
Illustration of the class imbalance problem. In anomaly detection, class imbalance occurs when the quantity of normal data points greatly surpasses that of anomaly data points. This imbalance poses challenges for both model training and performance assessment. Particularly, when anomaly data are scarce, the model may struggle to differentiate between normal and anomaly instances.

**Figure 6 sensors-24-05616-f006:**
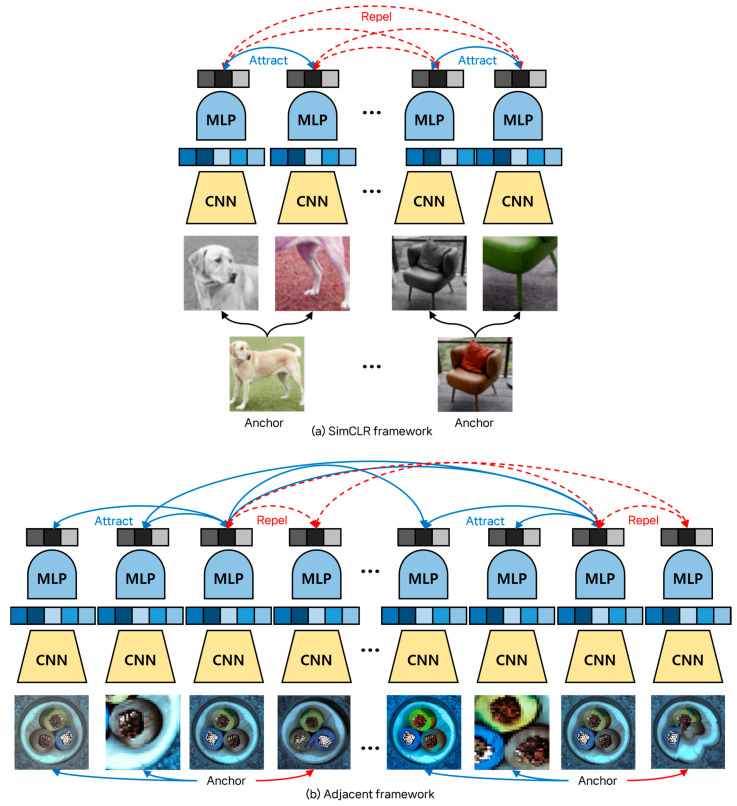
The difference between (**a**) SimCLR framework and (**b**) adjacent framework. While the SimCLR framework designates the training data within the batch as negative pairs, the adjacent framework pairs them as positive pairs. Notably, the adjacent framework embeds the features of normal data into the hypersphere space, resulting in improved discrimination between the features of normal data and those of anomaly data.

**Figure 7 sensors-24-05616-f007:**
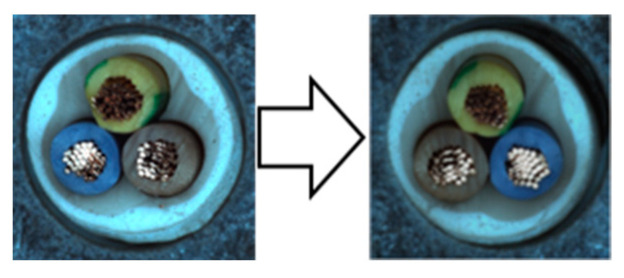
Depicted here is an image with Weak Overall augmentation. Weak Overall augmentation involves subtle adjustments to the anchor’s size and a mild application of Gaussian blur. Additionally, horizontal flipping occurs randomly with a specific probability. These Weak Overall samples aid in reducing sensitivity to minor overall changes.

**Figure 8 sensors-24-05616-f008:**
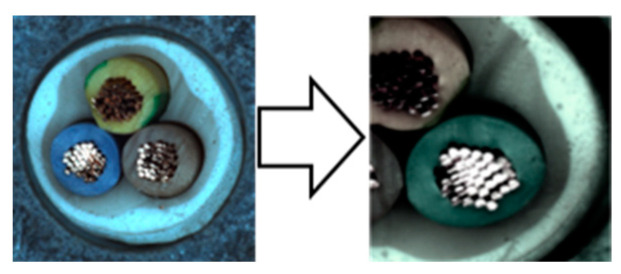
Depicted here is an image with Strong Overall augmentation. Strong Overall augmentation significantly alters the size and color of anchor images. Moreover, Gaussian blur, horizontal flipping, and grayscale are applied with varying probabilities. Strong Overall samples promote the learning of intricate features within normal images.

**Figure 9 sensors-24-05616-f009:**
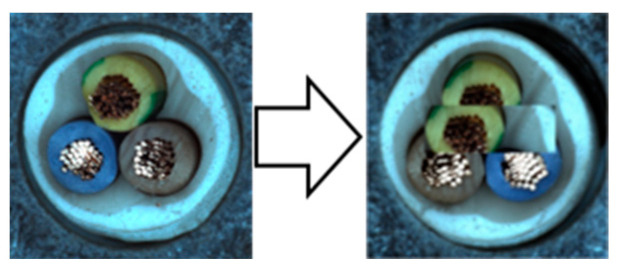
Depicted here is an image with CutPaste augmentation. CutPaste augmentation entails cutting a square patch from the anchor image and pasting it onto the original image. These CutPaste samples, which distort continuous patterns in normal images, facilitate the learning of discontinuous features present in anomaly data.

**Figure 10 sensors-24-05616-f010:**
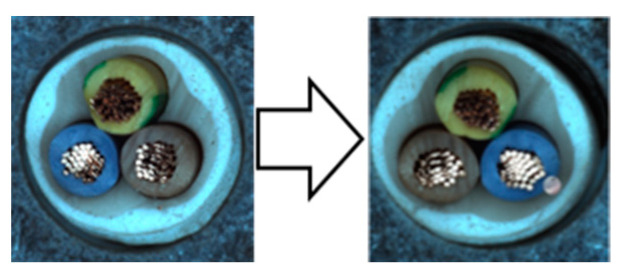
Depicted here is an image with SmoothBlend augmentation. SmoothBlend augmentation involves cutting a small, round patch from the anchor image and pasting it onto the original image. These SmoothBlend samples, which distort local detailed patterns in normal images, encourage the learning of detailed features found in anomaly data.

**Figure 11 sensors-24-05616-f011:**
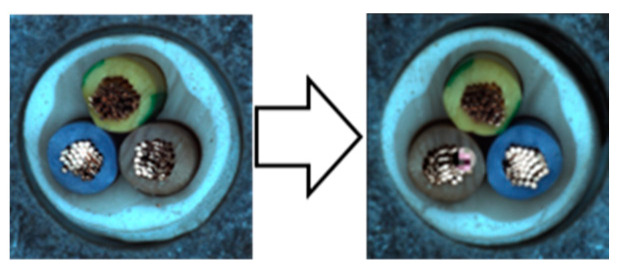
Depicted here is an image with Mosaic (ζ = 20) augmentation. Mosaic augmentation transforms color and resolution by specifying circular areas in anchor images. These Mosaic samples, which distort the resolution and color patterns of normal images, encourage the learning of natural and small defects present in anomaly data.

**Figure 12 sensors-24-05616-f012:**
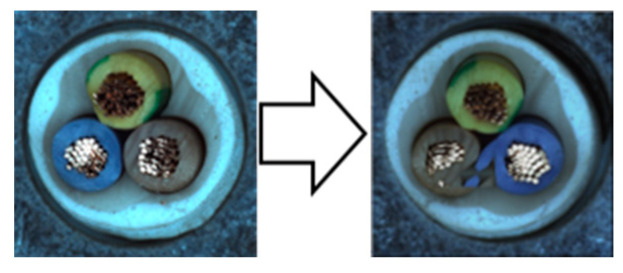
Depicted here is an image with Liquify (η = 0.03) augmentation. Liquify augmentation randomly selects a point on the training image and transforms its contours as they move. These Liquify samples maintain the shape of the normal image while distorting the contours, facilitating the learning of unnatural contours present in anomaly data.

**Figure 13 sensors-24-05616-f013:**
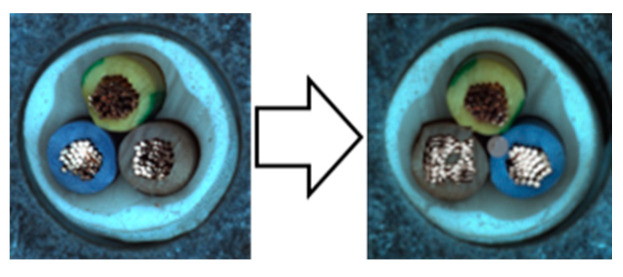
Depicted here is an image with Mosiquify augmentation. Mosiquify augmentation applies both Mosaic (ζ = 20) and Liquify (η = 0.03) augmentations to images. These Mosiquify samples, including two distorted anomalous patterns, promote the learning of various features from the anomaly images.

**Figure 14 sensors-24-05616-f014:**
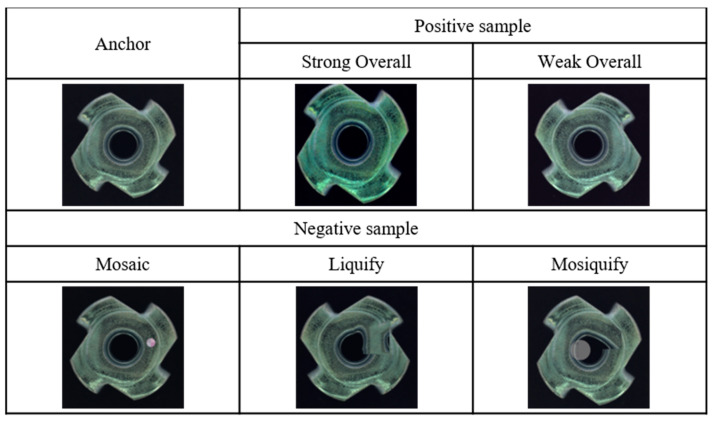
Presented here are images generated by adjacent augmentations. This figure showcases images created through adjacent augmentations, where Strong Overall augmentation and Weak Overall augmentation produce synthetic normal data, while Mosaic augmentation, Liquify augmentation, and Mosiquify augmentation generate synthetic anomaly data.

**Figure 15 sensors-24-05616-f015:**
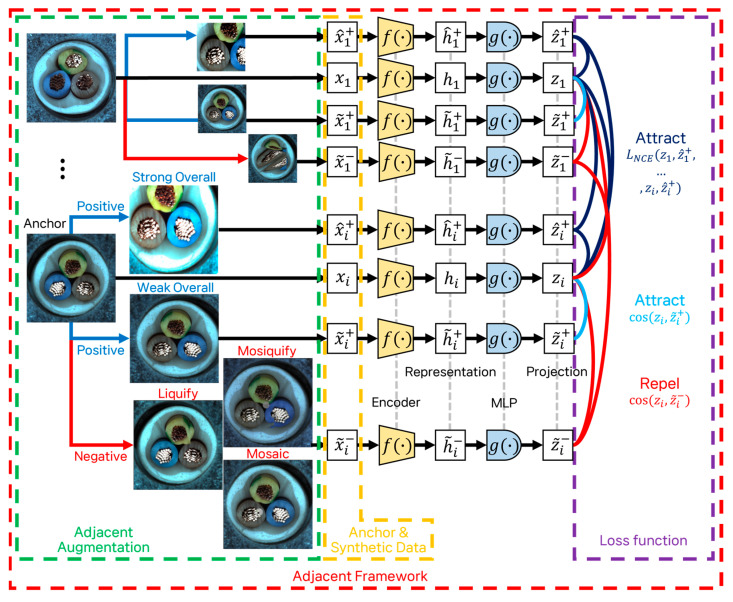
Overview of the adjacent augmentation and its framework. $x_{i}$: normal image serving as anchor. ${\hat{x}}_{i}$: positive sample generated by Strong Overall. ${\tilde{x}}_{i}^{+}$: positive sample generated by Weak Overall. ${\tilde{z}}_{i}^{-}$: negative sample generated by mimicking actual defects. The image passes through the encoder (f (∙)) to become a representation (h). The representation (h) passes through the projector (g (∙)), and then l2 normalization is applied to the projection (z).

**Figure 16 sensors-24-05616-f016:**
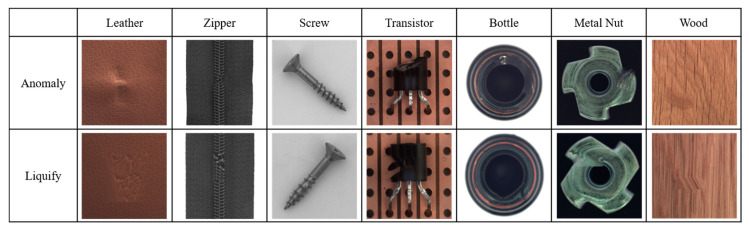
Comparison of real-world defects and synthetic anomaly images.

**Figure 17 sensors-24-05616-f017:**
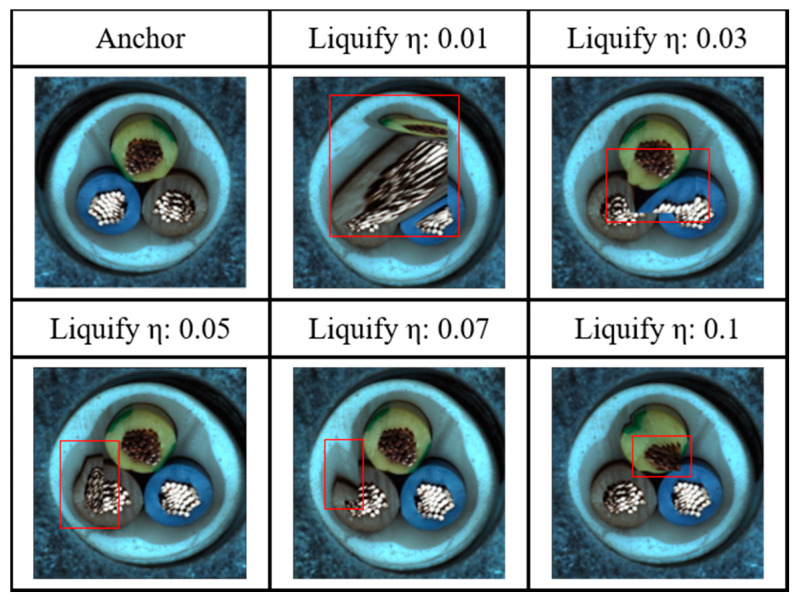
Liquify anomalous pattern size according to η.

**Table 1 sensors-24-05616-t001:** Comparison of augmentation speeds across various methods. This table presents the speeds of five augmentations: CutPaste, SmoothBlend, Mosaic, Liquify, and Mosiquify. Our proposed adjacent augmentation offers a simple augmentation approach with a speed comparable to previous methods.

	CutPaste [13]	SmoothBlend [14]	Mosaic	Liquify	Mosiquify
Millisecond	227.393	180.518	95.744	207.446	267.289

**Table 2 sensors-24-05616-t002:** Detailed information regarding the MVTec-AD dataset. This table outlines the quantity of normal images in the training set and the count of normal and anomaly images in the test set. Additionally, it specifies the number and types of defects within each category. Although this dataset represents an improvement over previous ones, there remains a shortage of anomaly data.

	Category	#Train	#Test (Good)	#Test (Defect.)	#Defect Groups	#Defect Regions	Image Side Length
Textures	Carpet	280	28	89	5	97	1024
	Grid	264	21	57	5	170	1024
	Leather	245	32	92	5	99	1024
	Tile	230	33	84	5	86	840
	Wood	247	19	60	5	168	1024
Objects	Bottle	209	20	63	3	68	900
	Cable	224	58	92	8	151	1024
	Capsule	219	23	109	5	114	1000
	Hazelnut	391	40	70	4	136	1024
	Metal Nut	220	22	93	4	132	700
	Pill	267	26	141	7	245	800
	Screw	320	41	119	5	135	1024
	Toothb.	60	12	30	1	66	1024
	Trans.	213	60	40	4	44	1024
	Zipper	240	32	119	7	177	1024
	Total	3629	467	1258	73	1888	-

**Table 3 sensors-24-05616-t003:** Differences in anomaly detection performance between SimCLR framework and adjacent framework (A.F.).

AU-ROC AU-PR	SimCLR [15] Mosaic ζ: 20	A.F. Mosaic ζ: 20	SimCLR [15] Liquify η: 0.05	A.F. Liquify η: 0.05	SimCLR [15] Mosiquify ζ = 20, η = 0.05	A.F. Mosiquify ζ = 20, η = 0.05
Framework and Aug. Category						
Zipper	0.789391 0.928752	0.773897 0.924250	0.829832 0.948985	0.942752 0.985565	0.788340 0.933215	0.738183 0.925600
Hazelnut	0.912143 0.954757	0.848929 0.911285	0.869286 0.932978	0.954286 0.975100	0.842143 0.908097	0.962500 0.979674
Bottle	0.917460 0.975251	0.998413 0.999500	0.986508 0.995740	1.000000 1.000000	0.938095 0.980658	0.950794 0.981918

**Table 4 sensors-24-05616-t004:** Maximum Area Under the Receiver Operating Characteristics (AU-ROC) and maximum Area Under the Precision-Recall (AU-PR) curves when applied with various augmentations.

AU-ROC AU-PR	CutPaste [13]	SmoothBlend [14]	Mosaic ζ = 20	Liquify	Mosiquify ζ = 20, η = 0.05
Aug. Category					
Leather	0.713315 0.889585	0.830163 0.942820	0.853940 0.944785	0.906590 0.967224 η: 0.01	0.635870 0.860233
Zipper	0.884979 0.964362	0.764968 0.929153	0.773897 0.924250	0.942752 0.985565 η: 0.05	0.738183 0.925600
Screw	0.923140 0.974795	0.819840 0.932187	0.792785 0.919263	0.927649 0.975877 η: 0.1	0.724739 0.892260
Hazelnut	0.863929 0.925698	0.916786 0.958774	0.848929 0.911285	0.954286 0.975100 η: 0.05	0.962500 0.979674
Tile	0.871573 0.942547	0.823232 0.920362	0.936869 0.976487	0.876263 0.952038 η: 0.01	0.898990 0.964675
Transistor	0.800417 0.763719	0.781250 0.706233	0.849167 0.813358	0.888750 0.877592 η: 0.1	0.772083 0.766140
Bottle	0.929365 0.978421	0.973810 0.992562	0.998413 0.999500	1.000000 1.000000 η: 0.05	0.950794 0.981918
Metal nut	0.886608 0.973357	0.804008 0.950171	0.838710 0.958632	0.927175 0.981952 η: 0.05	0.871457 0.966081
Toothbrush	0.672222 0.854802	0.880556 0.952743	0.758333 0.899457	0.894444 0.958447 η: 0.01	0.819444 0.930904
Wood	0.785088 0.927378	0.879825 0.964844	0.868421 0.959927	0.908772 0.971883 η: 0.1	0.858772 0.957199

**Table 5 sensors-24-05616-t005:** Ablation study of negative samples.

AU-ROC AU-PR	None	Mosaic ζ = 20	Liquify	Mosiquify ζ = 20, η = 0.05
Neg. Sample Category				
Leather	0.697351 0.869883	0.853940 0.944785	0.906590 0.967224 η: 0.01	0.635870 0.860233
Zipper	0.867122 0.965440	0.773897 0.924250	0.942752 0.985565 η: 0.05	0.738183 0.925600
Screw	0.876819 0.961209	0.792785 0.919263	0.927649 0.975877 η: 0.1	0.724739 0.892260
Hazelnut	0.925714 0.964005	0.848929 0.911285	0.954286 0.975100 η: 0.05	0.962500 0.979674
Tile	0.797619 0.906408	0.936869 0.976487	0.876263 0.952038 η: 0.01	0.898990 0.964675
Transistor	0.641250 0.543602	0.849167 0.813358	0.888750 0.877592 η: 0.1	0.772083 0.766140
Bottle	0.869048 0.954794	0.998413 0.999500	1.000000 1.000000 η: 0.05	0.950794 0.981918
Metal Nut	0.773216 0.940585	0.838710 0.958632	0.927175 0.981952 η: 0.05	0.871457 0.966081
Toothbrush	0.830556 0.930351	0.758333 0.899457	0.894444 0.958447 η: 0.01	0.819444 0.930904
Wood	0.715789 0.878648	0.868421 0.959927	0.908772 0.971883 η: 0.1	0.858772 0.957199

**Table 6 sensors-24-05616-t006:** Relationship between the size of the Liquify anomalous pattern and anomaly detection.

AU-ROC AU-PR	Liquify η: 0.01	Liquify η: 0.03	Liquify η: 0.05	Liquify η: 0.1
Liquify (η) Category				
Leather	0.906590 0.967224	0.866168 0.942729	0.861753 0.954633	0.791440 0.926733
Tile	0.876263 0.952038	0.855700 0.944713	0.858225 0.951333	0.825758 0.927673
Toothbrush	0.894444 0.958447	0.844444 0.943854	0.816667 0.930320	0.802778 0.914638
Zipper	0.849002 0.953625	0.847952 0.951083	0.942752 0.985565	0.877363 0.954699
Hazelnut	0.904286 0.943691	0.927143 0.958012	0.954286 0.975100	0.896071 0.941236
Carpet	0.519663 0.836309	0.508026 0.812877	0.731942 0.919088	0.676164 0.895256
Bottle	0.865873 0.952921	0.996825 0.999755	1.000000 1.000000	0.994444 0.998320
Metal Nut	0.822092 0.956493	0.817204 0.952127	0.927175 0.981952	0.810850 0.949913
Cable	0.794978 0.871893	0.846130 0.909616	0.802849 0.874685	0.847639 0.917611
Screw	0.726173 0.893507	0.703423 0.884927	0.782332 0.917255	0.927649 0.975877
Pill	0.633115 0.907203	0.729951 0.928439	0.655210 0.899663	0.735952 0.936548
Transistor	0.844167 0.823265	0.834583 0.818871	0.815000 0.804524	0.888750 0.877592
Wood	0.818421 0.939274	0.868421 0.960217	0.811404 0.942707	0.908772 0.971883
Grid	0.765246 0.900952	0.794486 0.920962	0.782790 0.920473	0.812865 0.921557
Capsule	0.735939 0.932523	0.768249 0.939492	0.799761 0.952384	0.814120 0.954620

**Table 7 sensors-24-05616-t007:** Maximum Area Under the Receiver Operating Characteristics (AU-ROC) and maximum Area Under the Precision-Recall curve (AU-PR) when various augmentation is applied in the Visual Anomaly (VisA) dataset [14].

AU-ROC AU-PR	CutPaste [13]	SmoothBlend [14]	Mosaic ζ = 20	Liquify η: 0.01	Mosiquify ζ = 20, η: 0.05
Method Category					
Pipe_fryum	0.869000 0.926897	0.917400 0.955720	0.874600 0.940887	0.958800 0.979326	0.899800 0.953652

**Table 8 sensors-24-05616-t008:** Compare Liquify and various anomaly detection methods.

AU-ROC	Avg.	Bottle	Cable	Capsule	Carpet	Grid	Hazeln.	Leather
Category Method								
GeoTrans [31]	67.2	74.4	78.3	67.0	43.7	61.9	35.9	84.1
GANomaly [32]	76.2	89.2	75.7	73.2	69.9	70.8	78.5	84.2
SPADE [33]	85.5	-	-	-	-	-	-	-
Liquify	88.3	100 η: 0.05	84.8 η: 0.1	88.6 η: 0.03	73.2 η: 0.05	81.3 η: 0.1	95.4 η: 0.05	90.7 η: 0.01
PaDiM [26]	95.3	-	-	-	-	-	-	-
PatchCore [27]	99.1	100	99.5	98.1	98.7	98.2	100	100
**AU-ROC**	**Metal nut**	**Pill**	**Screw**	**Tile**	**Toothb.**	**Trans.**	**Wood**	**Zipper**
**Category** **Method**								
GeoTrans [31]	81.3	63.0	50.0	41.7	97.2	86.9	61.1	82.0
GANomaly [32]	70.0	74.3	74.6	79.4	65.3	79.2	83.4	74.5
SPADE [33]	-	-	-	-	-	-	-	-
Liquify	92.7 η: 0.05	73.6 η: 0.1	92.8 η: 0.1	87.6 η: 0.01	89.4 η: 0.01	88.9 η: 0.1	90.9 η: 0.1	94.3 η: 0.05
PaDiM [26]	-	-	-	-	-	-	-	-
PatchCore [27]	100	96.6	98.1	98.7	100	100	99.2	99.4

## Data Availability

The data will be made available upon request.

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
