# Peer review of "Adjacent Image Augmentation and Its Framework for Self-Supervised Learning in Anomaly Detection"

_sensors, 2024, doi:10.3390/s24175616_

Round 1

Reviewer 1 Report (New Reviewer)

Comments and Suggestions for Authors

This paper proposes an adjacent augmentation technique that generates synthetic anomaly images, preserving object shapes while distorting contours to enhance correlation for self-learning algorithm. The comments are as follows. 

1. The novalty of the paper is small, as some data augmentaion methods describied in this paper are general. 

2. Some descriptions should be improved, for example:
(1) Why figure 1 is presented? it is not the main contribution of the paper. 

(2) It is supposed that MVTec-AD Dataset should be descrbed in experiments as it is a public dataset and not proposed in this paper. 

(3) The algorithm 1 is hard to be understanded. 

(4)  What's the contrastive learning framework mentioned in Line 85?

Author Response

comments 1: Why figure 1 is presented? it is not the main contribution of the paper. 
Thank you for pointing this out. The reason we presented Figure 1 was to demonstrate the diversity of contrastive learning and to clarify that we also used a similar method. I agree with this comment. Therefore, we have removed Figure 1.

comments 2: It is supposed that MVTec-AD Dataset should be described in experiments as it is a public dataset and not proposed in this paper. 
Thank you for pointing this out. We describe the MVTec-AD dataset in Chapter 2, Related Work, Section 2.1: MVTec-AD Dataset. This dataset also has a class imbalance issue.

comments 3: The algorithm 1 is hard to be understanded. 
Thank you for pointing this out. Algorithm 1 augments one anchor with three samples (Weak Overall sample, Strong Overall sample, and Negative sample). The anchor and positive samples are embedded closer together using NCE loss and cosine similarity loss. The anchor and negative sample are embedded farther apart using cosine similarity loss.

comments 4: What's the contrastive learning framework mentioned in Line 85?
Thank you for pointing this out. Our contrastive learning framework focuses on learning effective representations by embedding similar (positive) samples closer together in a latent space, while pushing dissimilar (negative) samples farther apart. Specifically, we augment each anchor image with both positive samples (such as Weak Overall and Strong Overall augmentations) and negative samples (synthetic anomaly images). We utilize losses like NCE loss and cosine similarity loss to ensure that positive pairs are closely aligned and negative pairs are distinct within the feature space. This approach not only improves the model's ability to distinguish between normal and anomalous data but also addresses class imbalance by leveraging synthetic anomaly images.

Reviewer 2 Report (New Reviewer)

Comments and Suggestions for Authors

This paper proposes an adjacent enhancement technology and a self-supervised learning framework to address the issue of class imbalance in anomaly detection. The experimental results show that this method achieves excellent performance on the MVTec-AD dataset, and the impact of anomaly pattern size on detection performance is also explored. In my opinion, this method effectively improves the accuracy and robustness of anomaly detection through innovative technical means, providing a reliable solution for practical applications. However, the following issues need to be addressed first:

1.    The introduction of the proposed adjacent enhancement technology in the article is clear, but more reasons for choosing the Mosaic, Liquify, and Mosiquify methods need to be provided to strengthen the argument.

2.    Although the article mentions comparisons with CutPaste and SmoothBlend, there is no in-depth discussion of the limitations or shortcomings of these existing methods. More emphasis should be placed on the advantages of the new method compared to these previous methods.

3.    In the experimental section, when "subsequently augmented" is mentioned, which specific data augmentation methods were used?

4.    The article mentions that "adjacent augmentations have contours like real-world defects". Could you please elaborate on how these augmentation methods generate abnormal samples? How are these samples used during training? What are the advantages of this generation method compared to other generation methods?

Comments on the Quality of English Language

Minor editing of English language required

Author Response

Comments 1: The introduction of the proposed adjacent enhancement technology in the article is clear, but more reasons for choosing the Mosaic, Liquify, and Mosiquify methods need to be provided to strengthen the argument.
Response 1: Thank you for your valuable feedback. We selected the Mosaic, Liquify, and Mosiquify methods based on their effectiveness in generating synthetic anomaly images that closely resemble real-world defects. These methods introduce realistic variations that challenge the model's ability to distinguish between normal and anomalous data, which is crucial for enhancing anomaly detection performance. Additionally, these techniques allow us to simulate a wide range of defects, addressing the class imbalance issue by providing diverse and realistic anomaly samples. Furthermore, these methods are particularly effective because they exploit the strong correlation between anomalous patterns and surrounding pixels, enabling more effective learning. We believe that these methods strengthen our framework by improving the model's robustness and generalization capabilities.

Comments 2: Although the article mentions comparisons with CutPaste and SmoothBlend, there is no in-depth discussion of the limitations or shortcomings of these existing methods. More emphasis should be placed on the advantages of the new method compared to these previous methods.
Response 2: Thank you for your insightful comments. We acknowledge the need for a more in-depth discussion of the limitations of existing methods like CutPaste and SmoothBlend, as well as a clearer comparison with our proposed approach. CutPaste and SmoothBlend are effective in generating synthetic anomalies, but they primarily focus on simple cut-and-paste operations or blending techniques, which may not fully capture the complexity of real-world defects. These methods often lack the capability to simulate intricate anomaly patterns that exist in diverse industrial settings. Additionally, they can sometimes introduce unrealistic artifacts that may hinder the model's generalization ability. In contrast, our proposed methods—Mosaic, Liquify, and Mosiquify—create more complex and realistic synthetic anomalies that better resemble real-world defects. By focusing on both local and global image distortions, our methods effectively simulate a broader range of anomaly types. Furthermore, our adjacent framework leverages the correlation between anomalous patterns and surrounding pixels, leading to more robust learning and better detection performance. This enhancement allows our method to overcome the limitations of previous approaches, improving the model’s ability to generalize across different types of anomalies and ultimately resulting in more accurate anomaly detection.

Comments 3: In the experimental section, when "subsequently augmented" is mentioned, which specific data augmentation methods were used?
Response 3: Thank you for your question. When we mentioned "subsequently augmented," we were referring to the data augmentation methods we proposed in our framework. Specifically, the weak overall augmentation and strong overall augmentation methods were used to generate positive samples, while the mosaic, liquify, and mosiquify augmentation methods were employed to create negative samples. These augmentations are integral to our approach, as they help the model effectively distinguish between normal and anomalous data.

Comments 4: The article mentions that "adjacent augmentations have contours like real-world defects". Could you please elaborate on how these augmentation methods generate abnormal samples? How are these samples used during training? What are the advantages of this generation method compared to other generation methods?
Response 4: Thank you for your insightful questions. When we refer to "adjacent augmentations," we are encompassing the Weak Overall, Strong Overall, Mosaic, Liquify, and Mosiquify methods. The detailed explanation of how these augmentation methods generate abnormal samples is provided in Chapter 3. To summarize, these augmentation methods simulate real-world defect patterns by introducing controlled distortions that resemble actual anomalies. For instance, as shown in Figure 15, positive samples generated through adjacent augmentations are embedded closer to the anchor using NCE loss and cosine similarity loss. In contrast, negative samples generated by the Mosaic, Liquify, and Mosiquify methods are embedded farther from the anchor using cosine similarity loss. The advantage of our generation method lies in its ability to produce a wide range of realistic anomalies that closely mimic real-world defects. This enhances the model's ability to distinguish between normal and anomalous data, making it more robust compared to other generation methods that might only focus on simpler or less varied synthetic defects.

This manuscript is a resubmission of an earlier submission. The following is a list of the peer review reports and author responses from that submission.

Round 1

Reviewer 1 Report

Comments and Suggestions for Authors

Technically, this paper is good and will be useful for readers. However, it needs to improve to be published in Sensors. It is suggested that the manuscript should be rewritten according to the comments.

1. In the 2.2 section, the autoencoder-based methods usually utilize the characteristic that anomaly data cannot be accurately reconstructed for anomaly detection. Why do these methods need to require accurate reconstruction of anomaly data?

2. Please provide more detailed explanations for Fig. 4.

3. There are some structural problems with introducing the methods in Chapter 3.

a. The division of chapters is confusing.

b. The introduction of the overall framework is not clear enough. The algorithm pseudocode needs to be supplemented in the manuscript。

4. Please provide more information about the experimental parameter settings of the model in Chapter 4.

5. The article mentions that the proposed method is a Self-Supervised Learning method. How does this Self-Supervised Learning work?

6. The introduction of the experimental section on comparing the baseline model is not given.

Author Response

Comments 1: In the 2.2 section, the autoencoder-based methods usually utilize the characteristic that anomaly data cannot be accurately reconstructed for anomaly detection. Why do these methods need to require accurate reconstruction of anomaly data?
Response 1: Thank you for pointing out the autoencoder. We agree that there was a lack of explanation. Therefore, We have revised it to the following sentence.
Autoencoder-based methods perform anomaly detection by reconstructing compressed input data as normal data. During the training phase, the model learns by repeatedly compressing and reconstructing normal data. In the inference phase, the model calculates the reconstruction error between the input data and the reconstructed data. Since the autoencoder reconstructs normal data well, the error is low, and the model classifies it as normal data. Conversely, the autoencoder does not reconstruct anomaly data well, resulting in a high error, and the model classifies it as anomaly data.
Page 4, Section 2.2 Representative Anomaly Detection, Second paragraph.

Comments 2: Please provide more detailed explanations for Fig. 4.
Response 2: Thank you for pointing out Fig. 4. We agree that there was a lack of explanation. Therefore, We have revised it to the following sentence.
This figure compares l2-autoencoder and SSIM-autoencoder for anomaly detection using autoencoders. An autoencoder trained on normal data compresses input fabric textures and then reconstructs them as normal fabric textures. The l2-autoencoder removes defects during reconstruction, while the SSIM-autoencoder retains defects. Therefore, the SSIM-autoencoder shows better anomaly detection performance than the l2-autoencoder.
Page 5, Section 2.2 Representative Anomaly Detection, Figure 4 caption.

Comments 3: There are some structural problems with introducing the methods in Chapter 3.
Comments 3-a: The division of chapters is confusing.
Response 3-a: Thank you for pointing out the structure of Chapter 3. We have revised the structure of Chapter 3 to 3.1 augmentation and 3.2 framework, and added an explanation of the structure.
This section describes the augmentation used in our framework and augmentation methods from previous work. The first two methods involve augmentation that generates positive samples, while the rest involve augmentation that generates negative samples.
Page 7, Chapter 3 methods.
Comments 3-b: The introduction of the overall framework is not clear enough. The algorithm pseudocode needs to be supplemented in the manuscript.
Response 3-b: Thank you for pointing out about the algorithm pseudocode. We agree with the inexistence of the algorithm pseudocode. therefore, we have added the algorithm pseudocode.
Page 12, Chapter 3 methods, Algorithm 1.

Comments 4: Please provide more information about the experimental parameter settings of the model in Chapter 4.
Response 4: Thank you for pointing out the experimental parameter settings. We agree with the deficiency in the experimental parameter settings. Therefore, we have added additional experimental parameter configurations.
We used the NVIDIA GeForce RTX 2080 Ti GPU.
Page 12, Chapter 4 Experiments, line 3.
The hyperparameters required for the experiments were applied in the same way as in the Spot-the-Difference [14] experiments.
Page 12, Chapter 4 Experiments, line 4.
Additionally, we used the Cosine Annealing Learning Rate method to gradually decrease the optimizer's learning rate following a cosine curve.
Page 12, Chapter 4 Experiments, line 6.
The cosine period is set according to the batch size, with the minimum learning rate set to 0, and the index of the last epoch at the start of training is set to -1.
Page 12, Chapter 4 Experiments, line 7.
Finally, ζ is a parameter that controls the size of mosaic anomaly patterns, and η is a parameter that controls the size of liquify anomaly patterns.
Page 12, Chapter 4 Experiments, line 12.

Comments 5: The article mentions that the proposed method is a Self-Supervised Learning method. How does this Self-Supervised Learning work?
Response 5: Thank you for pointing out Self-Supervised Learning. We agree that additional explanation about self-supervised learning is necessary. Therefore, we have added an explanation about self-supervised learning.
This framework employs a self-supervised learning method known as contrastive learning.
Page 10, Section 3.2, line 2.
In this approach, artificial labels are generated from the data to train the model, allowing for the effective utilization of unlabeled data.
Page 10, Section 3.2, line 3.
The framework involves increasing the similarity between positive pairs and decreasing the similarity between negative pairs.
Page 10, Section 3.2, line 4.

Comments 6: The introduction of the experimental section on comparing the baseline model is not given.
Response 6: Thank you for pointing out the comparison with the baseline model. We used the CutPaste [13] and SPD [14] as baselines, and we present the results in Table 4.
Page 13, Chapter 4 Experiments, Table 4.

Reviewer 2 Report

Comments and Suggestions for Authors

This work proposes a method for data augmentation by synthesizing anomaly images in order to deal with class imbalance issue. The method termed adjacency augmentation. To prove the efficacy of the framework, the authors used a ResNet50 network trained on the MVTec-AD dataset.

1-    I am very concerned that the authors achieved perfect results with an area under the roc curve of 100%, which suggest that the model returns perfect predictions (100% sensitivity, 100% specificity). No matter what the method is, it is not possible to achieve a perfect score, since there will be always a shift between training and testing set. This needs clarification.

2-    Please define abbreviation at first use: AU-ROC, AU-PR …etc.

3-    Could the authors confirm that the training and testing sets were completely separated?

4-    From the paper, it seems that the testing set is rather a validation set. If that is the case, the authors need to use a hold out test set for a fair evaluation.

5-    The authors need to include the learning curves.

6-     Please include the ROC curves for the results and the confusion matrix.

7-    How did you optimize the hyperparameters? What was the stopping criteria? More details in training and optimization are needed.

8-    The discussion section is completely missing. Please add a discussion section and discuss your findings, put them in the context of other works, include and discuss the impact of the deep learning architecture choice in the results https://doi.org/10.1002/nbm.4794                                

Comments on the Quality of English Language

Overall, the English is fine.

Author Response

Comments 1: I am very concerned that the authors achieved perfect results with an area under the roc curve of 100%, which suggest that the model returns perfect predictions (100% sensitivity, 100% specificity). No matter what the method is, it is not possible to achieve a perfect score, since there will be always a shift between training and testing set. This needs clarification.
Response 1: Thank you for pointing out the Area Under the ROC Curve (AU-ROC). As described in Section 2.1, the training set of the MVtec-AD dataset consists of normal images, while the test set is composed of a mix of different normal and anomaly images that are not present in the training set. Anomaly detection models are trained on the training set and evaluated on the test set. The anomaly detection method PatchCore achieves 100% Area Under the ROC Curve (AUROC) in the Bottle, Hazelnut, and Leather categories, as shown in Table 7.

Comments 2: Please define abbreviation at first use: AU-ROC, AU-PR …etc.
Response 2: Thank you for pointing out the abbreviation. We agree that defining abbreviations is necessary. Therefore, we have defined abbreviations upon their first use.
Experiments show that adjacency augmentation captures high-quality anomaly features, achieving superior Area Under the Receiver Operating Characteristics (AU-ROC) and Area Under Precision Recall (AU-PR) scores compared to previous methods.
Page 1, Abstract, line 5.
To reduce sensitivity to these weak overall changes, we use the Weak Overall augmentation from the Spot-the-Difference method.
Page 7, Section 3.1.1 Weak Overall, line 2.
We report the maximum Area Under the Receiver Operating Characteristics (AU-ROC) and Area Under Area Under Precision Recall (AU-PR) for each category in the MVTec-AD dataset.
Page 13, chapter 4 Experiments, line 11.

Comments 3: Could the authors confirm that the training and testing sets were completely separated?
Response 3: Thank you for pointing out the training and testing sets. The dataset for anomaly detection, as described in Section 2.1, consists of a training set composed of normal images, while the test set contains a mix of different normal images compared to those in the training set, along with anomaly images.

Comments 4: From the paper, it seems that the testing set is rather a validation set. If that is the case, the authors need to use a hold out test set for a fair evaluation.
Response 4: Thank you for pointing out the testing sets. As stated in previous responses, we did not use the validation set as the test set. We trained our model on the training set of the MVTec-AD dataset and tested it on the test set.

Comments 5: The authors need to include the learning curves.
Response 5: Thank you for pointing out the learning curves. We agree that learning curves are necessary. Therefore, we have added learning curves in Appendix A1.
Page 18, Figure A1.

Comments 6: Please include the ROC curves for the results and the confusion matrix.
Response 6: Thank you for pointing out the ROC curves and confusion matrix. We agree that ROC curves and confusion matrices are necessary. Therefore, we have added ROC curves in Appendix A2 and confusion matrix in Appendix A3.
Page 18, Figure A2, Figure A3.

Comments 7: How did you optimize the hyperparameters? What was the stopping criteria? More details in training and optimization are needed.
Response 7: Thank you for pointing out the issue with optimization. We agree that the explanation of optimization was insufficient. Therefore, we have revised the section related to optimization.
The hyperparameters required for the experiments were applied in the same way as in the Spot-the-Difference [14] experiments.
Page 12, Chapter 4 Experiments, line 4.
Additionally, we used the Cosine Annealing Learning Rate method to gradually decrease the optimizer's learning rate following a cosine curve.
Page 12, Chapter 4 Experiments, line 6.
The cosine period is set according to the batch size, with the minimum learning rate set to 0, and the index of the last epoch at the start of training is set to -1.
Page 12, Chapter 4 Experiments, line 7.

Comments 8: The discussion section is completely missing. Please add a discussion section and discuss your findings, put them in the context of other works, include and discuss the impact of the deep learning architecture choice in the results https://doi.org/10.1002/nbm.4794
Response 8: Thank you for pointing out the disscussion section. I agree with the absence of a discussion section. Therefore, we have included comparisons with other works, described the limitations of our approach, and discussed the impact of the choice of deep learning architecture in the discussion section.
Page 17, Chapter 5 Disscussion

Reviewer 3 Report

Comments and Suggestions for Authors

In this paper, the class imbalance problem in anomaly detection is proposed by the adjacency enhancement technique. It reduces the class imbalance by generating synthetic high-quality anomaly image features, which retain the shape of the object while distorting the outline to enhance the relevance to the surrounding environment. The adjacent-enhancement method achieved higher AU-ROC and AU-PR scores compared to previous methods, and the authors' model achieved perfect AU-ROC and AU-PR scores of 100% in the bottle category of the MVTec-AD dataset, improving the overall accuracy of the anomaly detection system. Some of the statements in this paper are ambiguous, and some problems need further revision and explanation.

1.      What influence does the comparison of the speed of adjacency enhancement methods in Table 1 have on adjacency enhancement techniques? The author can explain in detail.

2.       In the experimental part of Section 4, only the MVTec-AD dataset was validated. Can this method also reduce class imbalance in other datasets?

3.       In Table 3, "Comparison of various enhancement methods based on adjacency Framework and SimCLR framework", there is no comparison result of adjacency enhancement Mosiquify method mentioned in section 3 above.

4.      The AUROC and AU-PR scores of the Liquify method used by the author in Table 7 are lower than those of the PatchCore method in other anomaly detection methods, which the author can explain in detail.

Comments on the Quality of English Language

I was satisfied with the quality of the English expressions.

Author Response

Comments 1: What influence does the comparison of the speed of adjacency enhancement methods in Table 1 have on adjacency enhancement techniques? The author can explain in detail.
Response 1: Thank you for pointing out the speed of adjacency enhancement methods. We agree that the explanation of optimization was insufficient. Therefore, we have revised the section related to optimization.
Our proposed adjacency augmentation offers a simple augmentation approach with a speed comparable to previous methods.
Page 2, Table 1 caption, line 3

Comments 2: In the experimental part of Section 4, only the MVTec-AD dataset was validated. Can this method also reduce class imbalance in other datasets?
Response 2: Thank you for pointing out the other datasets. We agree that validation on other datasets is necessary. Therefore, we have added Table 7 to validate the Pipe_fryum category of the Visual Anomaly (VisA) dataset.
Page 16, Table 7.

Comments 3: In Table 3, "Comparison of various enhancement methods based on adjacency Framework and SimCLR framework", there is no comparison result of adjacency enhancement Mosiquify method mentioned in section 3 above.
Response 3: Thank you for pointing out the Table 3. We agree that there are no results for Mosiquify in Table 3. Therefore, we have supplemented Table 3 with the experimental results for Mosiquify.
Page 13, Table 3

Comments 4: The AUROC and AU-PR scores of the Liquify method used by the author in Table 7 are lower than those of the PatchCore method in other anomaly detection methods, which the author can explain in detail.
Response 4: Thank you for pointing out the Table 7. We agree that there are no results for Mosiquify in Table 3. Therefore, we have supplemented Table 3 with the experimental results for Mosiquify.
Page 18, Section 5.4. Limitations.

Round 2

Reviewer 2 Report

Comments and Suggestions for Authors

After reviewing the responses, I am very concerned about the methodology and the results. The authors trained a neural network on normal images only, but the model is able to detect anomalies with a perfect AUC! this contradicts state-of-the-art papers.

From the learning curves provided, there is no validation curve but only training! how did you choose the best model? this suggests an overfitting.

By confusion matrix, I meant to show the results in a confusion matrix, not its definition.

Comments on the Quality of English Language

Moderate editing.